# Expression of Human PTEN-L in a Yeast Heterologous Model Unveils Specific N-Terminal Motifs Controlling PTEN-L Subcellular Localization and Function

**DOI:** 10.3390/cells8121512

**Published:** 2019-11-26

**Authors:** Teresa Fernández-Acero, Eleonora Bertalmio, Sandra Luna, Janire Mingo, Ignacio Bravo-Plaza, Isabel Rodríguez-Escudero, María Molina, Rafael Pulido, Víctor J. Cid

**Affiliations:** 1Departamento de Microbiología y Parasitología, Facultad de Farmacia, Universidad Complutense de Madrid (UCM) & Instituto Ramón y Cajal de Investigaciones Sanitarias (IRYCIS). Plaza de Ramón y Cajal s/n, 28040 Madrid, Spain; teresafe@farm.ucm.es (T.F.-A.); eleonora.bertalmio@edu.unito.it (E.B.); igbravo93@gmail.com (I.B.-P.); isabelre@farm.ucm.es (I.R.-E.); molmifa@ucm.es (M.M.); 2Instituto de Investigación Sanitaria Biocruces Bizkaia, 48903 Barakaldo, Spain; Sandra.LunaBlanco@osakidetza.eus (S.L.); JANIRE.MINGOGOMEZDEOTEO@osakidetza.eus (J.M.); 3IKERBASQUE, Fundación Vasca para la Ciencia, 48011 Bilbao, Spain

**Keywords:** PTEN, phosphoinositides, subcellular localization, alternative translation initiation, *Saccharomyces cerevisiae*, PI3K, heterologous expression

## Abstract

The tumour suppressor PTEN is frequently downregulated, mutated or lost in several types of tumours and congenital disorders including PHTS (PTEN Hamartoma Tumour Syndrome) and ASD (Autism Spectrum Disorder). PTEN is a lipid phosphatase whose activity over the lipid messenger PIP_3_ counteracts the stimulation of the oncogenic phosphatidylinositol 3-kinase (PI3K) pathway. Recently, several extended versions of PTEN produced in the cell by alternative translation initiation have been described, among which, PTEN-L and PTEN-M represent the longest isoforms. We previously developed a humanized yeast model in which the expression of PI3K in *Saccharomyces cerevisiae* led to growth inhibition that could be suppressed by co-expression of PTEN. Here, we show that the expression of PTEN-L and PTEN-M in yeast results in robust counteracting of PI3K-dependent growth inhibition. N-terminally tagged GFP-PTEN-L was sharply localized at the yeast plasma membrane. Point mutations of a putative membrane-binding helix located at the PTEN-L extension or its deletion shifted localization to nuclear. Also, a shift from plasma membrane to nucleus was observed in mutants at basic amino acid clusters at the PIP_2_-binding motif, and at the Cα2 and CBR3 loops at the C2 domain. In contrast, C-terminally tagged PTEN-L-GFP displayed mitochondrial localization in yeast, which was shifted to plasma membrane by removing the first 22 PTEN-L residues. Our results suggest an important role of the N-terminal extension of alternative PTEN isoforms on their spatial and functional regulation.

## 1. Introduction

PTEN (Phosphatase and tensin homologue deleted on chromosome 10) is one of the most prominent tumour suppressor genes in mammalian cells. This key function is attributable to the dephosphorylation of phosphatidylinositol(3,4,5)P_3_ (PIP_3_), a minor phosphoinositide species produced from phosphatidylinositol(4,5)P_3_ (PIP_2_) at the plasma membrane (PM) by class I phosphatidylinositol 3-kinase (PI3K) in response to extracellular stimuli. PIP_3_ functions as the key second messenger for activation of the Akt protein kinase, which governs multiple functions related to tumorigenesis. As the major PIP_3_ phosphatase, PTEN antagonizes PI3K-Akt-dependent signaling to counteract cell proliferation, inhibition of apoptosis, and glycolysis stimulation, among other PIP_3_-dependent processes [1,2,3,4]. In addition, besides its predominant role as a lipid phosphatase, PTEN function as a protein phosphatase is presumed on some protein targets, often involved in cancer, such as focal adhesion kinase (FAK), beta-catenin, insulin receptor substrate (IRS1), platelet-derived growth factor (PDGF) receptor, or protein tyrosine kinase 6 (PTK6), among others [5,6,7]. Dysfunction of PTEN is related to several human diseases. A plethora of somatic mutations, either within coding or non-coding regions of the gene, as well as loss of heterozygosity, have been found in many types of cancer, i.e., glioblastoma, prostate, mammary gland, and endometrium tumors. In addition, germline mutations have been found in inherited PTEN hamartoma tumor syndromes (PHTS), namely Cowden and Bannayan-Riley-Ruvalcaba, as well as in autism spectrum disorders (ASD) [8,9].

PTEN was initially described to encode a 403 amino acids/55 kDa protein belonging to the superfamily of protein tyrosine phosphatases (PTPs) and dual specificity protein phosphatases (DSPs), characterized by the presence of the catalytic signature motif HCXXGXXR [10]. However, it was also noted that it presented weak phosphatase activity on phosphopeptides, while it could accommodate acidic and larger substrates, such as lipids, due to the presence of basic residues in the phosphatase domain and to a wider catalytic pocket [11,12,13]. The PTEN protein folds into two major lobes: the phosphatase fold and a C2 domain that facilitates its attachment to the PM. Besides, its N-terminal end contains a PIP_2_-binding motif (PBM) overlapping with a nuclear localization signal (NLS) and its C-terminal tail harbours a PDZ domain involved in protein-protein interactions [14,15,16,17,18]. Within the phosphatase domain, Cys124 acts as the essential residue for the catalysis of substrate dephosphorylation, and the C124S mutant is usually used as a fully catalytically-inactive PTEN enzyme. The Gly129 residue, also in the active pocket, is essential for lipid phosphatase activity, but a G129E mutant, unlike C124S, retains protein phosphatase activity [12]. The fact that both mutations have the same penetrance in pathological phenotypes indicates that dephosphorylation of PIP_3_ is the crucial function of PTEN in disease development.

PTEN cellular function and subcellular localization are regulated through a large number of post-translational modifications and protein-protein interactions [1,4,19,20]. For example, PTEN mono-ubiquitination favours its accumulation within the nucleus, where it can regulate chromosomal stability and cell cycle regulation [21,22,23]; and phosphorylation of its C-terminal tail gives rise to a conformational change that makes the protein catalytically inactive and promotes its dissociation from the PM [24,25,26]. Moreover, PTEN is able to form homodimers with greater lipid phosphatase activity than that of the monomers, and the dimerization of a disease-related mutant with a wild type version of PTEN may have a dominant-negative effect on PTEN function [27,28]. Specific poly-ubiquitination regulates negatively PTEN dimerization, PM recruitment and function [29]. Understanding the complex regulation of PTEN is essential to devise preventive and therapeutic strategies against PTEN-associated malignancies and syndromes.

In addition to this intricate regulation, PTEN protein was recently unveiled to exist within the cell as alternative translational isoforms [30,31]. Upstream and in-frame of the canonical AUG translation-initiation site in the unusually long PTEN mRNA, there is a non-AUG site (CUG) that gives rise to an alternative longer isoform containing an N-terminal extra extension of 173 amino acids, called PTEN-L (also known as PTENα) [32]. In virtue of putative secretion and import signals in such extension, PTEN-L can be secreted out of the cell, enter recipient cells and reduce PIP_3_-dependent signaling or even tumour size in mice xenografts [30,33]. In addition, PTEN-L is also targeted to the mitochondria, where it regulates energy metabolism [31]. More recently, the other three PTEN proteoforms have been identified, resulting from additional non-AUG initiation sites downstream that of PTEN-L, AUU, CUG and CUG, which account, respectively, for PTEN-M (146 amino acids extra; also known as PTENβ), PTEN-N (131 amino acids extra) and PTEN-O (72 amino acids extra) [34]. We are only starting to understand the physiological significance of these alternative PTEN translational variants, but it is assumed that they may play important roles on the spatial regulation of PTEN and its interaction with other proteins. All N-terminal extended PTEN versions display a putative membrane binding helix (MBH) that has been proposed to reinforce PTEN membrane tethering to allow proximity to its substrate [34,35], as well as to bind parkin (PRKN) at the mitochondrial outer membrane to regulate mitophagy [36,37]. In contrast, PTEN-M has been reported to localize mostly in the nucleolus, where it dephosphorylates nucleolin [38].

We have used the model eukaryotic organism S. cerevisiae to reconstitute the PI3K-PTEN mammalian pathway by heterologous expression of its main elements [39,40,41,42]. Our model has been extensively used for the analysis of pathogenic PTEN variants [43,44]. This system relies on the toxicity exerted by a PM-targeted mammalian class I PI3K when overexpressed in the yeast cell, due to the conversion of essential PM PIP_2_ pools into PIP_3,_ which is naturally absent and lacks functional significance in this organism. The co-expression of wild type PTEN with PI3K counteracts this lethality, whereas catalytically inactive or loss-of-function PTEN mutants are not able to rescue yeast growth in the presence of PI3K. This experimental setting provides an easy readout of PTEN competence as a PIP_3_ phosphatase in vivo. In our model, the expression of a version of PTEN-L lacking its N-terminal secretion signal efficiently reverted PI3K-derived toxicity [39]. Here, we use PTEN-L fusions to GFP to study in the yeast model, the determinants of PTEN-L localization and function, mapping intrinsic domains and residues that are involved in PTEN-L mitochondrial localization and in the balance between its PM attachment and its accumulation in the nucleus.

## 2. Materials and Methods

### 2.1. Media and Strains

YPD [1% (*w*/*v*) yeast extract, 2% (*w*/*v*) peptone and 2% (*w*/*v*) dextrose] broth or agar was the general non-selective medium used for yeast cell growth. Synthetic medium (SD) contained 2% glucose, 0.17% yeast nitrogen base without amino acids, 0.5% ammonium sulphate and 0.12% of the synthetic amino acid drop-out mixture, lacking appropriate amino acids and nucleic acid bases to maintain selection for plasmids. For SG (synthetic galactose) and SR (synthetic raffinose) media, glucose was replaced with 2% (*w*/*v*) galactose or 1.5% (*w*/*v*) raffinose, respectively. *GAL1-*driven protein induction in liquid medium was performed by growing cells in SR to mid-exponential phase and then refreshing the cultures to an OD600 of 0.3 directly with SG or with SR supplemented with galactose 2% (SR-Gal) for 5–8 h. Growth assays on plates were performed as described in Rodriguez-Escudero et al. [40].

*S. cerevisiae* strain YPH499 (*MATa ade2-101 trp1-63 leu2-1 ura3-52 his3-200 lys2-801*) was the strain used for all experiments, unless noted. The strain VHY87 (*MATα leu2-3, 112 ura3-52 his4 can1R TRP1::DsRed-HDEL*), was a generous gift of M. Cyert (Stanford University, CA, USA). *Escherichia coli* DH5α F’[K12D(*lacZYA-argF*)U169 *deoR supE44 thi-1 recA1 endA1 hsdR17 gyrA96 relA1* (w*80lacZ*D*M15*)F´] was used for routine molecular biology techniques.

### 2.2. Plasmids

Transformation of *E. coli* and yeast and other basic molecular biology methods were carried out using standard procedures.

pYES2-PTEN (amino acids 1-403) and pYES2-PTEN-L* (amino acids 22-L-576-L; amino acid nomenclature according to Pulido [32]) have been previously described [39], pYES2-PTEN-L (amino acids 1-L-576-L) was generated by PCR adding to PTEN-L* the N-terminal 21 residues, pYES2-PTEN-M.1 and pYES2-PTEN-M.2 (amino acids 28-L-576-L) were constructed by mutagenic PCR from pYES2-PTEN-L*. pYES2-GFP-PTEN-L, pYES2-GFP-PTEN-M and pYES2-GFP-PTEN-L* were constructed by amplifying GFP with the primers GFP-PTEN-L-fw (CCAAGCTTATGAGTAAAGGAGAAGAA) and GFP-PTEN-L-rv (CCAAGCTTTTTGTATAGTTCATCCATGC), both designed with *Hin*dIII flanking sites (underlined) and subsequent cloning into pYES2-PTEN-L, pYES2-PTEN-M or pYES2-PTEN-L* [39]. pYES2-PTEN-L-GFP and pYES2-PTEN-L*-GFP were constructed by subcloning PTEN-L and PTEN-L* (with *Hin*dIII/*Bgl*II flanking sites) into pYES2-GFP.

pYES2-GFP-PTEN-L*-∆MBH, pYES2-GFP-PTEN-L*-∆R6, GFP-PTEN-L-KRR-AAA, pYES2-GFP-PTEN-L*-Cα2 and pYES2-GFP-PTEN-L-CBR3 were all constructed by overlapping PCR. The strategy implied the amplification of the N-terminal part of PTEN-L with the general forward primer PTEN-L *Bam*HI (CGGATCCATGTCTGAGTCGCCTGT) and a reverse primer containing the desired mutation, and of the C-terminal part with a complementary forward primer with the mutation and the general reverse primer PTEN-L *Eco*RI (CCGAATTCTCAGACTTTTGTAATTTGTG). The overlapping PCR was then digested with and subcloned into *Bam*HI and *Eco*RI sites at pYES2-GFP-PTEN-L*. Internal primers containing the mutations were: ∆MBH-rv (CTCTTTGATGATGGCGGTACCCTGCAGGATGGAAATGGCTC) and ∆MBH-fw (CATTTCCATCCTGCAGGGTACCGCCATCATCAAAGAGATCGTTA), accounting for a substitution of the residues between K151-L and M174-L by a single G residue; KRR-AAA-rv(CCTCTTGATATGCAGCTGCGTTTCTGCTAACGATCTCTT) and KRR-AAA-fw (TAGCAGAAACGCAGCTGCATATCAAGAGGATGGATTCG), accounting for the mutations K13A/R14A/R15A (according to short canonical PTEN numbering); CBR3-rv (TGCGTCGGCTGCTCCAGCTGCGTTCTGTTTGTGGAAATGTTTCACTTTTGGGTAAATAGAAC) and CBR3-fw (GCAGCTGGAGCAGCCGACGCAATGTTTCACTTTTGGGTAAATA), accounting for the mutations 263-269 KMLKKDK-AAGAADA (according to short canonical PTEN numbering); Cα2-rv (TGCGTTGGCTGCGTCTGCACCAGCTGCGTCAAGATCATTTTTTGTTAAAG) and Cα2-fw (GCAGCTGGTGCAGACGCAGCCAACGCATACTTTTCTCCAAATTTTAAG), accounting for the mutations 327–335 KANKDKANR-AAGADAANA (according to short canonical PTEN numbering).

By PCR site-directed mutagenesis on pYES2-GFP-PTEN-L* we introduced the following mutations that are named according to the numbering of PTEN-L [L162A-L (PTEN-L-L/A), L158A-L/L159A-L (PTEN-L-LL/AA), and L158A-L/L159A-L/L162A-L (PTEN-L-LLL/AAA)], or to the conventional numbering of PTEN (C124S, K13A, K13E, K13N, K13Q, K13T, R14A, R14G, R14M, R14S, R15G, R15I, R15K, R15S, and K13A/R14A/R15A). The mutations I28A-L and M174I-L were also introduced by PCR site-directed mutagenesis. All primers are available upon request.

The mitochondrial marker Ilv6-mCherry, encoded in the plasmid YEplac112-Ilv6-mCherry, was constructed by amplifying 280 bp of the *ILV6* promoter region followed by its coding sequence with primers containing *Sph*I-*BamH*I flanking sites (ACAT**GCATGC**TGAGTGCTTATCTAGGATG and CG**GGATCC**ACCAGGTGGTAGTTGGG), and further subcloning into YEplac112-mCherry [45].

### 2.3. Fluorescence Microscopy and Image Analysis

For in vivo fluorescence microscopy (GFP and/or mCherry observation), cells that were exponentially growing in SR-Gal for 5 h were harvested by centrifugation at 5000 rpm for 1 min and viewed directly. Cells were examined with an Eclipse TE2000U microscope (Nikon) using the appropriate sets of filters. Digital images were acquired with Orca C4742-95-12ER charge-coupled device camera (Hamamatsu, Japan) and were processed with the HCImage software (Hamamatsu, Japan).

Fiji software [46] was used for image analysis. Two different parameters were used to reflect the variations in the distribution of GFP-PTEN-L along an individual yeast cell: the PM/cytoplasm relative fluorescence intensity (RFI) ratio and the nucleus/cytoplasm RFI ratio. RFIs were calculated by using the ROI manager tool in order to measure fluorescence at a selected region of pixels. The PM/cytoplasm RFI was calculated by dividing the average fluorescence intensity of three different small areas along the PM of a single yeast cell by that of a selected area inside its cytoplasm. This is a way of correcting the differences in raw intensities among the different cells. The nucleus/cytoplasm RFI was calculated by dividing the fluorescence intensity of a selected area inside the nucleus by that of the cytoplasm. Ten cells were included in the analysis for each experiment and the resulting average was used to calculate a final average and standard deviation of three biological replicates for each experiment. The percentage of cells with GFP-PTEN-L nuclear signal was based on the analysis of 50 cells of each clonal population. Statistical significance was determined with Student’s *t*-test.

### 2.4. Preparation of Cell Lysates and Immunodetection

Yeast extracts, SDS-PAGE and immunoblotting were performed as previously described [40].Primary antibodies used were anti-PTEN (C-Term)-clone Y184 (Cat. #04-409 Millipore^®^, Burlington, MA, USA), anti-GFP-JL8 (Cat. #632380, Living Colors^®^, Takara, Mountain View, CA, USA) and anti-G6PDH (Sigma, St. Louis, MO, USA; 1:50,000 dilution). Secondary antibodies were anti-mouse IgG-Alexa Fluor^®^ 680, anti-rabbit IgG-IRDye^®^ 800CW and anti-rabbit IgG-IRDye^®^ 680; all from LI-COR (Lincoln, NE, USA) at 1:5,000 dilution.

## 3. Results and Discussion

### 3.1. Expression of N-Terminal Extended PTEN Translational Variants in Yeast

We previously described that activity of PTEN PIP_3_-phosphatase activity is traceable in yeast [40,43]. The PTEN-L translational variant exhibits an N-terminal extension of 173 amino acids (Figure 1A). We have also reported that a version of PTEN-L lacking its first 22 residues (hereafter named PTEN-L*), thus excluding the cleavable N-terminal signal peptide involved in PTEN-L secretion in higher cells described by Hopkins et al. [30], maintained full activity in the yeast model [39]. To further explore the properties of N-terminally extended PTEN translational variants and to study their subcellular localization, we expressed in yeast a series of these variants—namely full length PTEN-L, PTEN-L* and PTEN-M, both fused and non-fused to GFP under the control of the galactose-inducible and glucose-repressible *GAL1* promoter. Canonical PTEN was also included for comparison (Figure 1B).

Immunodetection with an anti-PTEN pan antibody showed that PTEN and PTEN-L were expressed in low levels as compared to PTEN-L*, while PTEN-M levels were intermediate (Figure 2A). To understand whether the yeast model was able to recognize PTEN alternative initiation codons, we made two PTEN-M versions, one with an ATG codon at the start position (PTEN-M.1) and the other with the Ile-encoding alternative initiation codon (PTEN-M.2). Interestingly, both forms were similarly expressed, although the PTEN-M.2 version displayed a slower mobility. In contrast, changing the Met in position 1 of classic short PTEN, to Ile, led to total lack of expression (Figure 2A). This suggests that, as reported for higher cells, yeast is able to read alternative start codons in the PTEN-L mRNA, but the PTEN ATG is essential for expression of short canonical PTEN. Changing of Ile28-L to Ala (I28A-L), however, did not affect expression of PTEN-L or PTEN-L*, indicating that the artificial Met codon drives expression of these variants regardless of the alternative Ile28-L start codon (Figure 2A). PTEN-L* was expressed in higher levels than PTEN-L and -M. Functionally, the I28A-L PTEN-L mutant was less efficient rescuing PI3K-induced growth inhibition (Figure 2B), an effect that was not patent in PTEN-L*, likely due to its high expression levels. This suggests that the Ile28-L residue is important for the function of PTEN-L, but only when expression is limited.

Longer exposures for the immunoblots revealed that short canonical PTEN was also produced from the PTEN-L mRNA, as revealed by the appearance of a band of equivalent size to that expressed from the short PTEN cDNA that was missing when a point mutation in the PTEN start codon was introduced in the PTEN-L cDNA (PTEN-L M174I-L) (Figure 2C).

N-terminal GFP fusions of PTEN-L, PTEN-L* and PTEN-M expressed in high levels and gave rise to additional bands of higher mobility, detectable with anti-PTEN but not with the anti-GFP antibodies, some apparently corresponding in size to PTEN-L* or PTEN (Figure 2C), while other major bands present both in the absence and presence of N-terminally fused GFP, may correspond to proteolytic products. Thus, both N-terminal extensions and short canonical PTEN were produced from the GFP-PTEN-L, -L* and -M cDNAs. The amount of PTEN protein was comparable to that expressed from the short, non-extended PTEN cDNA. Mutation of the ATG translation initiation site of PTEN in the constructs either encoding PTEN (PTEN M1I), PTEN-L* (PTEN-L* M174I-L) or GFP-PTEN-L* (GFP-PTEN-L* M174I-L), led to the disappearance of the lowest Mw band (Figure 2C), confirming that it corresponded to canonical PTEN.

In sum, as reported in higher cells [31,34,38], at least two different translation initiation sites are recognized by the yeast initiation complexes on mRNAs encoding N-terminally extended PTEN isoforms: one is the first AUG after the 5´cap, artificially added in our constructs, and the other is the internal recognition site for the canonical PTEN. Thus, the different PTEN cDNAs expressed in yeast gave rise to the long and short PTEN isoforms simultaneously, even in the presence of additional N-terminal extensions, as the GFP-encoding frame.

### 3.2. N-Terminally Extended GFP-PTEN Variants Are Functional in Yeast and Localize to the Plasma Membrane and the Nucleus

Next, we systematically tested the ability of PTEN variants to counteract PI3K-induced growth inhibition in yeast, as read-out of their catalytic activity in vivo. As shown in Figure 3A,B, all PTEN variants tested rescued growth in the presence of the p110α PI3K catalytic subunit, demonstrating their functionality, with the exception of PTEN M1I, lacking the start ATG codon of the canonical short PTEN cDNA (Figure 3B), which was not expressed (see negative controls in Figure 2A,C). Unlike PTEN M1I, both PTEN-L* M174I-L and GFP-PTEN-L* M174I-L, in which expression of canonical short PTEN is abrogated, rescued PI3K-toxicity to the same extent as the respective wild type versions, and even more efficiently than canonical PTEN. This result confirms full catalytic competence for N-terminal extended PTEN versions and is consistent with previous work showing the effect of PTEN isoforms on Akt activation in higher cells [30,31,34,38].

The fact that anti-GFP antibody only recognized the fusion proteins (Figure 2C, right panel) endorsed that only these versions would be visualized in the cell by fluorescence microscopy, allowing to study their localization. GFP-PTEN-L, -L* and -M fusion proteins robustly localized to the yeast PM in all cells, although a much fainter cytoplasmic localization and nuclear enrichment were also observed (Figure 3C–D). Nuclear localization was confirmed by transforming the cells in a strain with the ER marker dsRed-HDEL, which labels the perinuclear ER, and thus surrounded the intracellular GFP signal emitted by GFP-PTEN-L (Figure 3C). We were not able to detect a GFP-PTEN fusion either by fluorescence microscopy or immunoblotting, suggesting that it cannot be stably expressed in yeast (data not shown). Nevertheless, this prominent PM localization of PTEN-L, PTEN-L* and PTEN-M contrasted with the nucleo-cytoplasmic distribution purported for canonical PTEN in higher cells [17,18]. This suggests that particular PM-tethering signals exist at the N-terminal region of these extended PTEN variants. Thus, we decided to perform site-directed mutagenesis on GFP-PTEN-L fusion proteins to investigate the contribution of different residues and motifs of PTEN-L to PM tethering vs. nuclear localization in the yeast heterologous model.

### 3.3. Clusters of Positively Charged and Hydrophobic Amino Acids at the C2 Domain Partially Contribute to PM-Nucleus Partition of GFP-PTEN-L in the Yeast Model

PTEN tertiary structure folds into two distinct domains: an N-terminal phosphatase domain and a C-terminal C2 domain [13]. C2 domains are usually involved in the interaction with membrane lipids in a Ca^2+^-dependent or -independent way. The PTEN C2 domain presents two different regions with an important role in positioning the enzyme towards the PM, the CBR3 and the Cα2 loops (Figure 1A), allowing the access of the phosphatase domain to its substrate. The positive charges in these regions favour the interaction of the PTEN C2 domain with lipids in a Ca^2+^-independent manner [13,47]. Lee *et al*. [13] designed a series of mutations in either the CBR3 or the Cα2 loop, in which the positive and/or hydrophobic residues that were topologically exposed towards the PM were changed to Gly or Ala, (i.e., in the CBR3 mutant, KMLKKDK^263–269^ was mutated to AAGAADA^263–269^; and in the Cα2 mutant, KANKDKANR^327–335^ was changed to AAGADAANA^327–335^). We reproduced these mutations in GFP-PTEN-L* to evaluate the contribution of these C2 loops to its robust PM localization. Both mutants displayed a slightly reduced activity as compared to wild type GFP-PTEN–L* (Figure 4A), although they were expressed as efficiently (Figure 4B). This is consistent with previous observations by Lee et al. on canonical PTEN, who reported that these mutants presented a reduced tumor suppressor activity [13], as well as our previous report that mutations in these C2 regions compromised PTEN function [41]. Nevertheless, these basic regions were not necessary, at least individually, to tether PTEN-L* to the PM, as GFP fluorescence was still enriched at the periphery of yeast cells (Figure 4C). However, both mutants showed a significantly diminished PM/cytoplasm RFI ratio, indicating that the ability of these mutant versions to attach to the PM was partially compromised (Figure 4D). Nuclear fluorescence in these GFP-tagged versions was clearly increased (Figure 4C), and consistently nucleus/cytoplasm RFI ratio was significantly enhanced (Figure 4E). The mutations at the CBR3 loop had a greater effect on both loss of PM signal and nuclear enrichment than those in the Cα2 loop. We derived two conclusions from these experiments. First, partially compromising PM tethering of PTEN-L* leads to enrichment of the protein in the nucleus; and second, PM recognition sites in the C2 domain are not individually responsible for robust localization of GFP-PTEN-L*. Thus, we were prompted to test the unique N-terminal extension of PTEN-L for its contribution to PM localization.

### 3.4. The Membrane Binding Helix at the N-Terminal Extension of PTEN-L Is Essential for Plasma Membrane Localization but not for Its Phosphatase Activity

Masson et al. proposed that a putative α-helix within PTEN-L extension (Membrane Binding Helix, MBH, consisting of residues 151-L to 174-L, corresponding the residue 174 to Met1 of canonical PTEN, Figure 1A) might favor its interaction with the PM in a more stable state than that of PTEN, resulting in an advantageous catalysis [35]. Therefore, we decided to analyze the function of the MBH in PTEN-L* localization in the yeast model. We mutated to Ala three hydrophobic residues, L158-L, L159-L and L162-L, which are conserved in the paralogous phosphatases TPTE and *Ci-VSP* [35]. These point mutations were introduced alone [L162A-L (PTEN-L*/L-A)] or in combination [L158A-L/L159A-L (PTEN-L*/LL-AA) and L158A-L/L159A-L/L162A-L (PTEN-L*/LLL-AAA)]. We also constructed a version lacking the whole MBH (PTEN-L*-∆MBH), which was substituted by a single Gly residue, thus missing as well Met1 of canonical PTEN at its translation initiation site (M174-L in PTEN-L). Neither mutation of Leu residues nor removal of the whole putative helix affected the phosphatase activity of PTEN-L (Figure 4A). This could not be attributed to the expression of canonical PTEN from the long GFP-PTEN-L* mRNAs, as the PTEN-L*-∆MBH form lacked the ATG codon so it could not produce this protein (Figure 4B). All versions reached an expression equivalent to that of the wild type GFP-PTEN-L* (Figure 4B).

Interestingly, although GFP signal at the PM for the single or multiple Leu to Ala mutants was comparable to that of WT GFP-PTEN-L*, it was totally absent for PTEN-L*-∆MBH. However, all these mutants, especially PTEN-L*/LLL-AAA and PTEN-L*-∆MBH showed a relocation to the yeast nucleus (Figure 4F–H). We counted at the fluorescence microscope the percentage of cells with apparent nuclear accumulation of GFP-PTEN-L*. For WT GFP-PTEN-L*, while all cells displayed the characteristic PM pattern, only 71% of these cells additionally showed detectable nuclear signal. In contrast, we found that 100% of cells from cultures expressing any of the MBH mutant proteins exhibited nuclear GFP signal, and the nucleus/cytoplasm RFI ratio of GFP-PTEN-L*almost doubled (in the L-A mutant) or tripled (in the LLL/AAA or ∆MBH mutants) that of the WT (Figure 4H). Our results imply that the MBH is indeed essential for robust PM localization of PTEN-L, as proposed by Masson et al. [35] and, furthermore, that it strongly prevents PTEN-L import into the nucleus. It is particularly intriguing that the GFP-PTEN-L* version deleted for the MBH, which does not express the short version of PTEN and is severely concentrated in the nucleus, is still functional, as judged for its ability to counteract PI3K toxicity in yeast (see Figure 4A). Thus, in spite of its effect on PM tethering, this putative helix does not seem to significantly contribute to PTEN-L activity on PIP_3_ dephosphorylation in vivo.

### 3.5. The Polyarginine Stretch in PTEN-L Is Necessary for Nuclear Localization

When Hopkins et al. [30] described PTEN-L for the first time, they realized the presence of a conserved poly-Arg sequence in its N-terminal extension (Figure 1A), which was involved in the ability of secreted PTEN-L to be internalized by acceptor cells. An independent study showed that this stretch was responsible for nucleolar localization of PTEN-M in HeLa cells [38]. We constructed a version lacking such sequence (PTEN-L* ∆R6) for its expression in yeast. GFP-PTEN-L* ∆R6 was expressed as efficiently as GFP-PTEN-L* (Figure 5A). Peculiarly, the major degradation band present in all other constructs was missing, suggesting that it is the product of Arg-directed protease cleavage at this site (Figure 5A, white arrowhead). GFP-PTEN-L* ∆R6 kept intact its localization to the yeast PM as compared to the WT (Figure 5B), displaying a PM/cytoplasm RFI ratio similar to that of GFP-PTEN-L* (Figure 5C). However, it completely lost its nuclear localization (Figure 5B). The absence of the poly-Arg stretch did not affect GFP-PTEN-L* activity in our model (Figure 5D). Thus, either translocation or retention of GFP-PTEN-L* into the yeast nucleus seems dependent of the presence this poly-Arg stretch. This result is in consonance with evidence reported by Liang *et al*., who found that the mutation of this sequence eliminated PTEN-M nucleolar localization [38], and with the in-silico prediction of the PTEN-L R6 motif as a nuclear localization sequence [48].

### 3.6. The Basic Cluster at the PIP_2_-Binding/NLS Region Regulates PTEN-L Localization at the PM and Nucleus

The N-terminus of canonical PTEN contains a PIP_2_-binding motif (PBM) overlapping a nuclear localization signal (NLS) and therefore accounts for the regulation of PTEN subcellular localization and activity [18,49]. Thus, this region is a hotspot for PTEN loss-of-function mutations. We previously reported that a cluster of basic amino acids within the PTEN PBM, Lys13, Arg14 and Arg15 (KRR^13–15^; positions 186-L to 188-L in PTEN-L; Figure 1A), is crucial for the PM/nucleus targeting of PTEN in mammalian cells, as well as for its phosphatase activity in vivo [17]. We wondered whether this motif was also necessary for PTEN-L distribution within the yeast cell, in addition to the above-described MBH. To this end, we introduced several point mutations in GFP-PTEN-L*, some of them previously studied in PTEN, either found in tumors (K13E, K13N, K13Q, K13T, R14M, R14G, R14S, R15I, R15G, R15K and R15S) or generated for functional analysis [(K13A, R14A and K13A/R14A/R15A (KRR-AAA)], and compared the behavior of these mutant versions to that of WT GFP-PTEN-L*. The expression of all these versions was similar to that of GFP-PTEN-L* (Figure 6A). None of the single point mutations tested caused a significant variation in the PM localization of GFP-PTEN-L* (Figure 6B) and the phosphatase activity of PTEN-L* was preserved (Figure 6C). However, the KRR-AAA mutant lost both PM localization and activity (Figure 6B-C). Interestingly, this implies that both the MBH in PTEN-L extension and the KRR motif at the PBM/NLS domain are necessary but not sufficient for stable tethering of PTEN-L versions to the PM.

Regarding nuclear localization, the substitution of the Lys13 residue (according to classic PTEN nomenclature, for simplicity; Lys186-L in PTEN-L) for Ala (K13A), Asn (K13N), or Thr (K13T) caused the total exclusion of GFP-PTEN-L* from the nucleus. On the contrary, its mutation to Asp (K13E) or Gln (K13Q) led to a significant increase in the percentage of cells showing a nuclear localization of GFP-PTEN-L* (Figure 6D). This suggests that this position can act as a pivot for PTEN-L nuclear localization. In the case of the Arg14 (Arg187-L in PTEN-L) and Arg15 (Arg188-L in PTEN-L) residues, all changes analyzed led to a significant enhancement of the percentage of cells with nuclear signal of GFP-PTEN-L* (Figure 6D). Such effect was also observed in the triple mutant KRR-AAA.

The lack of nuclear PTEN-L signal in some point mutants targeting Lys186-L (Lys13in PTEN-L) may reflect either a more robust attachment to the PM or the loss of a key determinant for nuclear import or retention. In this regard, we have reported that a PTEN K13R mutant shifts PTEN from the nucleus to the PM in mammalian cells [17]. Also, Nguyen et al. [50] found an enhanced association to the PM and a block of nuclear localization of a canonical short PTEN K13A mutant as compared to the WT. It is intriguing that other substitutions in the same residue behave as those in Arg187-L and Arg188-L, showing the opposite effect, that is, reinforcement of nuclear localization. This seems to indicate that Lys13/Lys186-L in PTEN is the core of a functional NLS both in canonical PTEN and in N-terminal extended versions. However, it is intriguing that the triple KRR-AAA mutant displays nuclear localization in spite of bearing a change to Ala in Arg186-L. Our previous mutational analyses on canonical short PTEN suggest that nuclear concentration of PTEN correlates with a decrease of a PIP_3_ phosphatase activity in mammalian cells [51]. However, PTEN-L does not seem to follow this rule in the yeast model, as several mutants described here in either the MBH or the KRR motifs caused a strong increase of PTEN-L* accumulation in the nucleus without affecting its function. Also, in our previous work, we reported that the PTEN R15K mutant enhanced nuclear accumulation in higher cells [17], in consistence with reports by other authors who showed that the Arg15 residue was essential for the PM targeting of PTEN [50]. Although our results on PTEN-L* support previous data on the short version of PTEN, the R188K-L mutation (R15K in PTEN) did not eliminate all GFP-PTEN-L* peripheral signal, likely because PM localization is reinforced by the MBH at the N-terminal extension.

Peculiarly, some of the oncogenic mutations that lead to a complete loss of function of PTEN phosphatase activity in the yeast model (K13E, R15I, R15S, R15K) [39,43], did not cause such effect on GFP-PTEN-L* (K186E-L, R188I-L, R188S-L, R188K-L). Actually, as discussed above, only changing all three residues in the KRR motif to Ala led to PTEN-L* dysfunction in yeast. Interestingly, the different behaviour of N-terminal extended PTEN and canonical PTEN may suggest that PTEN-L is either more robust as an enzyme or simply more stable than PTEN. Expression analyses, as revealed by immunoblots, consistently showed more abundance of PTEN-L* than PTEN in yeast (see Figure 2A,C, Figure 4B, Figure 5A and Figure 6A), favoring the notion that the N-terminal extension confers stability. Alternatively, structural determinants (such as the MBH) present in the N-terminal extension could compensate individual changes at the KRR motif. However, the KRR motif is essential whereas the MBH is dispensable for function. It is likely that the MBH confers an accessory mean to keep a stable interaction of PTEN-L with the PM, whereas the KRR motif, that is present in all PTEN isoforms, supports a more important role for the phosphatase activity of the protein. This view is consistent with previous evidence demonstrating that, while the allosteric activation of PTEN enzymatic activity is determined by the binding to PIP_2_ through the PBM [14], PTEN-L seems constitutively activated [52], yielding it less sensitive to single changes at the KRR motif.

### 3.7. The First 22 Amino Acids of PTEN-L Drive the Protein to Mitochondria, Precluding Plasma Membrane Localization

GFP-tagging of PTEN-L at its N-terminus might interfere with the function of motifs in such extension. Thus, we also produced PTEN-L and PTEN-L* fusions to GFP at the C-terminus. When detected by immunoblot with specific anti-PTEN and anti-GFP antibodies, we observed that PTEN-L-GFP versions were expressed less efficiently than GFP-PTEN-L in yeast (Figure 7A). This is in agreement with the idea that, as stated above, a bulky N-terminal extension may contribute to stabilize PTEN-L. Also, we noticed that PTEN-L-GFP was less abundant in yeast cell lysates than PTEN-L*-GFP (Figure 7A). Tzani et al. [34] also observed that PTEN-M was more abundant than PTEN-L in higher cells, suggesting that the putative N-terminal secretion signal in PTEN-L, missing in PTEN-L* and PTEN-M, may limit the expression or stability of the protein. The levels of expression of these versions are co-related with their ability to counteract PI3K-induced toxicity in yeast, as PTEN-L-GFP was less efficient than any other fusion as a PIP_3_ phosphatase in vivo (Figure 7B). A lower mobility band corresponding to the Mw of PTEN-GFP and detectable with both anti-GFP and anti-PTEN antibodies appears in PTEN-L-GFP and PTEN-L*-GFP lysates and is missing in the N-terminally tagged GFP-PTEN-L and GFP-PTEN*-L samples (Figure 7A) indicating that the short canonical PTEN is indeed translated in yeast from Met174-L (Met1 in PTEN) PTEN-L cDNAs. 

Regarding subcellular localization, PTEN-L*-GFP behaved like the equivalent N-terminal fusions (Figure 7C, left panel). However, full-length PTEN-L-GFP signal was poorer, consistent with the low expression observed by immunoblot, and the protein was not associated with the PM, but instead enriched in cytoplasmic spots (Figure 7C, right panel). Since PTEN-L has been described to be localized to mitochondria [37], we co-expressed an Ilv6-mCherry fusion to detect these organelles. As shown in Figure 7D, full-length PTEN-L-GFP, in contrast to PTEN-L*-GFP, lacking 22 amino acids at its N-terminus, was enriched in mitochondrial compartments. We had not observed this phenomenon with N-terminal GFP fusions described above, likely due to the fact that the fluorescent protein may have masked an essential mitochondrial localization motif. This result defines the existence of a mitochondrial localization signal in the first 22 residues of PTEN-L and highlights the role of different regions at the N-terminal extensions of PTEN variants on subcellular localization.

In summary, heterologous expression of N-terminally extended PTEN variants demonstrates that the yeast model is competent to produce both extended and short canonical PTEN isoforms from PTEN-L cDNA, even in the presence of longer 5′ extensions in the mRNA, such as the GFP in-frame gene fusion used here. Furthermore, our results support the hypothesis that PTEN-L displays a more robust performance in vivo over PTEN as a PIP3-phosphatase. Finally, our results suggest a determinant role of the N-terminal extension motifs in the subcellular localization of PTEN long isoforms. Namely, we show that its poly-Arg motif is important for nuclear localization, the MBH for attachment to the PM, and the first amino acids of PTEN-L for localization to mitochondria (see Figure 8 for a summary). Importantly, the observation of these phenomena in a heterologous model such as *S. cerevisiae*, implies that all these are intrinsic features of PTEN-L, in the absence of other human proteins.

## Figures and Tables

**Figure 1 cells-08-01512-f001:**
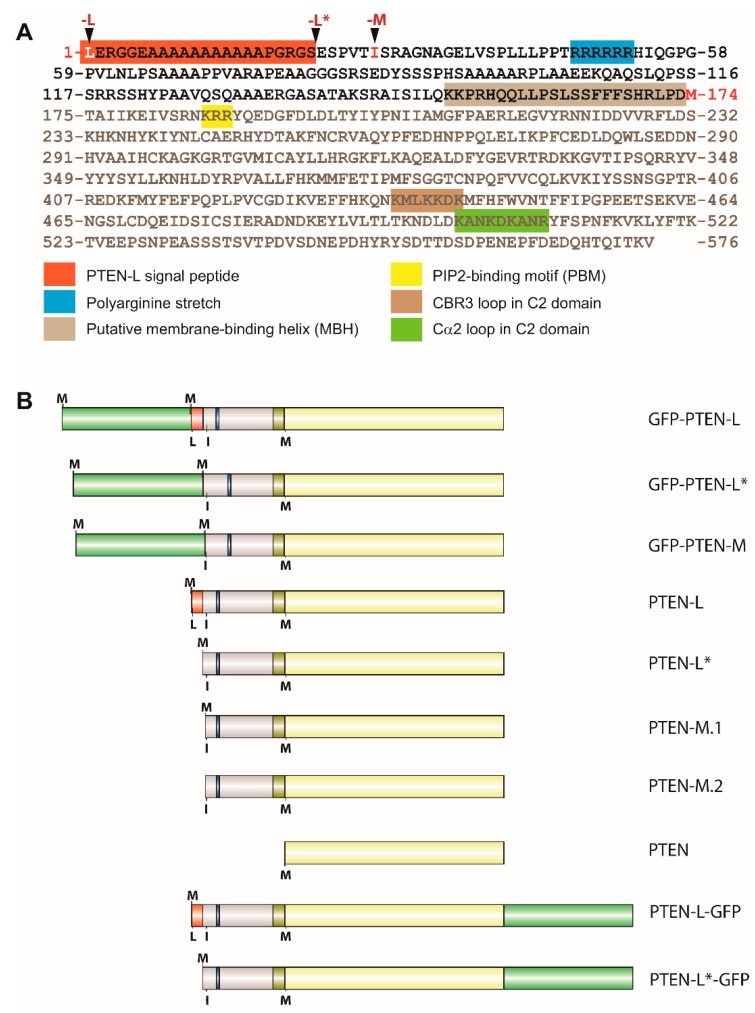
Primary structure of PTEN-L and constructs developed in this work for expression in S. cerevisiae. **A**. Amino acid sequence of PTEN-L marking the domains and motifs relevant for this work. Starting residues for coding sequences of PTEN-L, PTEN-M, and PTEN are highlighted (in white –PTEN-L– or red). The putative signal peptide, missing in our artificial PTEN-L* and PTEN-M constructs, is marked in orange, the poly-Arg stretch in blue, the putative membrane binding helix (MBH) in light brown, the Lys-Arg-Arg core of the PBD/NLS region in yellow, and the CBR3 and Cα2 loops within the C2 domain in brown and green respectively, as indicated. Amino acid numbering corresponds to PTEN-L (accession NP_001291646). **B**. Scheme of the versions of PTEN used in this work, indicating in the bottom of each depiction the canonical (M, methionine) or alternative (L, leucine; I, isoleucine) translation start codons. At the top of each depiction, the artificial M residues used to initiate the translation of some isoforms are indicated. GFP is represented in green, and the N-terminal signal peptide, poly-Arg stretch and MBH follow color codes as in A. All versions were expressed from the pYES2 yeast expression vector.

**Figure 2 cells-08-01512-f002:**
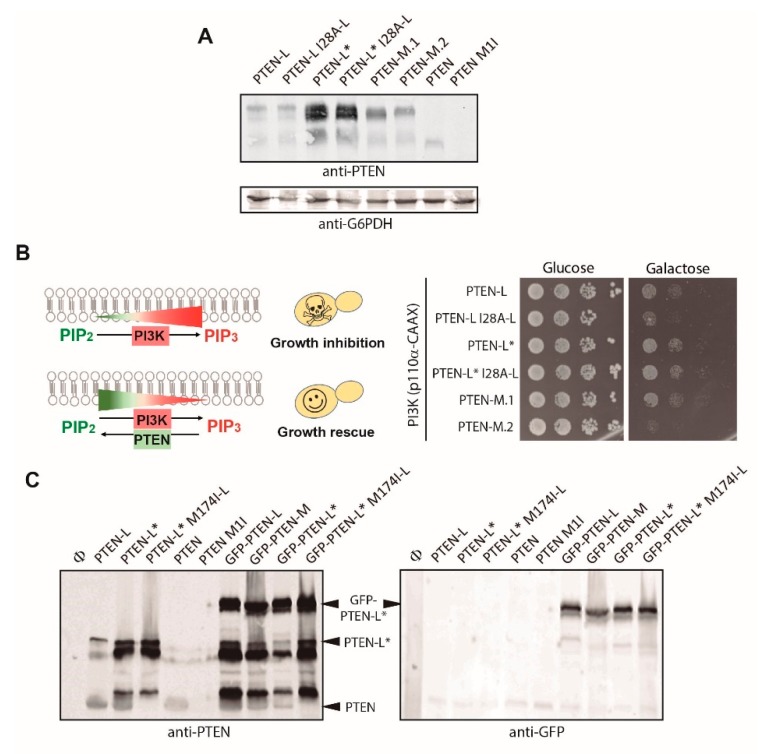
Expression in yeast of N-terminal extended PTEN variants. **A.** Immunoblots on lysates obtained from yeast transformants expressing the indicated versions of PTEN. The same membrane was hybridized with anti-PTEN antibodies (upper panel) and anti G6PDH (lower panel) as a loading control. **B.** Functional assay of PTEN versions as in A. The rationale for the yeast growth functional assay for PTEN in vivo activity is shown in the left panel. PI3K activity inhibits yeast growth by limiting the availability of the essential PIP_2_ phosphoinositide, which is turned into PIP_3_. Co-expression of PTEN recovers PIP_2_ levels, allowing growth. In the right panel, agar drop growth assays in repression conditions (Glucose, control, left) or in induction conditions (Galactose, right) on co-transformants bearing the indicated versions of PTEN and a growth-inhibiting version of PI3K (p110α-CAAX). SD Ura- Leu- plates were grown for 3 days at 28 °C. **C.** Expression of mutants lacking the canonical ATG start codon of PTEN (Met1 = Met174-L). Samples from lysates expressing the indicated forms of PTEN were immunoblotted with both anti-PTEN (left) and anti-GFP antibodies (right). Φ means empty vector.

**Figure 3 cells-08-01512-f003:**
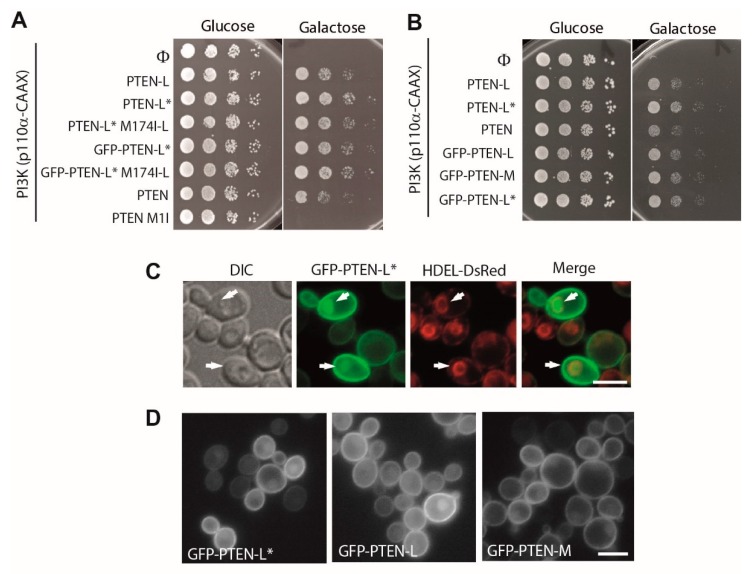
Function and subcellular localization of N-terminal extended PTEN variants in yeast. **A**. Functional assays of PTEN and GFP-PTEN versions lacking the canonical ATG start codon of PTEN (Met1 = Met174-L). **B.** Comparative functional assessment of GFP-PTEN-L and GFP-PTEN-M with GFP-PTEN-L* and control lacking GFP. **C.** Bright field Differential Interferential Contrast (DIC) and fluorescence microscopy of the same representative field of VHY87 strain (which expresses HDEL-DsRed for ER visualization) transformed with pYES2-GFP-PTEN-L*. **D.** Fluorescence microscopy on YPH499 cells expressing the indicated versions of GFP. Only the green channel is shown. Bars in C and D represent 5 µm.

**Figure 4 cells-08-01512-f004:**
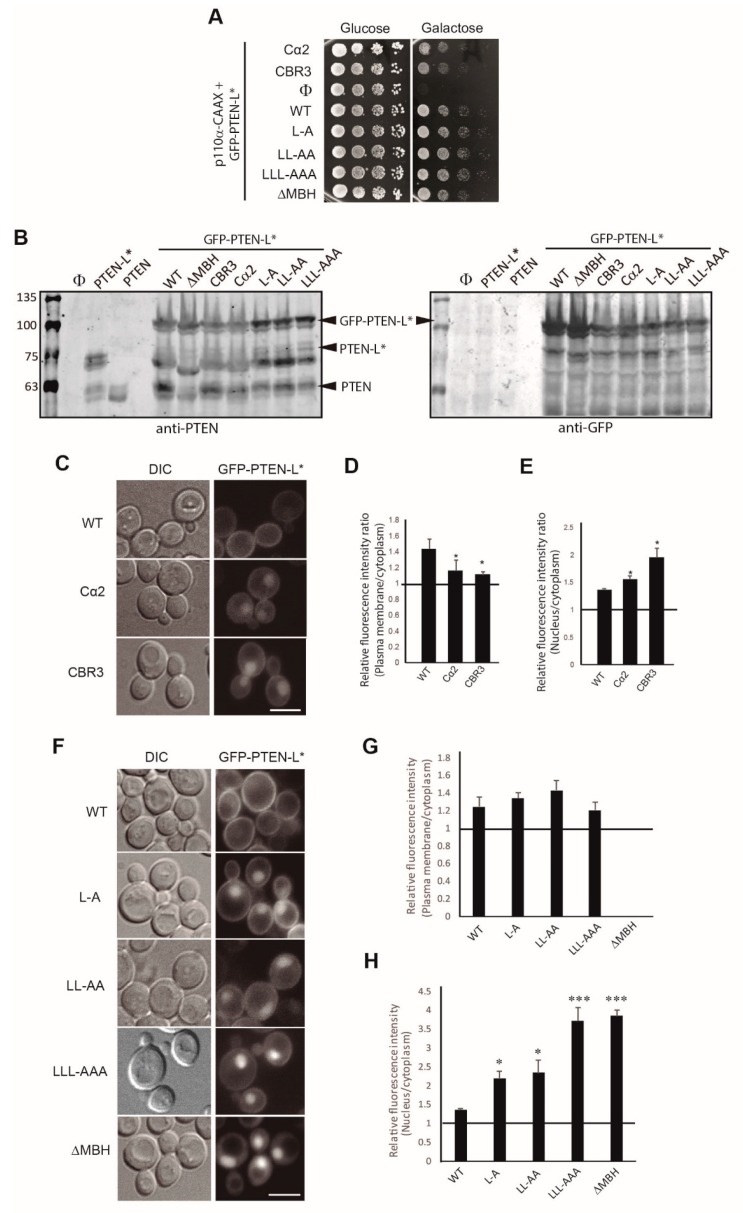
Localization and function of PTEN-L mutants targeting the C2 loops and the MBH. **A.** Agar drop growth assays on co-transformants bearing the indicated versions of GFP-PTEN-L* and p110α-CAAX. **B.** Immunoblots on lysates obtained from yeast transformants expressing the indicated versions of PTEN or GFP-PTEN-L*, as indicated. The same membrane was hybridized with anti-PTEN antibodies (left) and anti-GFP antibodies. Scale bars correspond to 5 µm. Error bars graphs correspond to the standard deviation (**p* < 0.05; ****p* < 0.005, according to Student’s *t*-test). **C.** Bright field (DIC) and fluorescence microscopy of YPH499 strain transformants expressing WT GFP-PTEN-L* or bearing multiple mutations (see text) in the Cα2 and CBR3 motifs, as indicated. Average plasma membrane:nucleus RFI ratios of transformants as in A. Average nucleus:cytoplasm RFI ratios of transformants as in A–B. **E.** Average nucleus:cytoplasm RFI ratios of transformants as in **C**–**D**. **F.** Bright field and fluorescence microscopy of YPH499 strain transformants expressing WT GFP-PTEN-L* or bearing single (L–A), double (LL–AA), triple (LLL–AAA) mutations (see text) or a deletion in the putative membrane-binding helix (MBH) in the N-terminal extension, as indicated. **G.** Average plasma membrane:cytoplasm RFI ratios of transformants as in F.

**Figure 5 cells-08-01512-f005:**
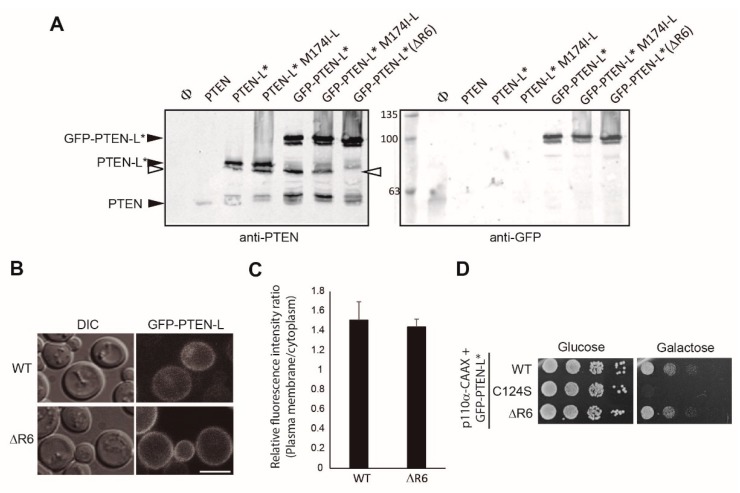
Expression in yeast of a GFP-PTEN-L* deleted in its poly-Arg stretch. **A.** Immunoblots on lysates obtained from yeast transformants expressing the indicated versions of PTEN or GFP-PTEN-L*, as indicated. The same membrane was hybridized with anti-PTEN (left) and anti-GFP antibodies. The white arrowheads point a band, likely a degradation product, that is present in all PTEN-L* forms except the ΔR6 mutant. **B.** Bright field (DIC) and fluorescence microscopy of YPH499 strain transformants expressing WT GFP-PTEN-L* or the ΔR6 mutant. Scale bar corresponds to 5 µm. **C.** Average plasma membrane: cytoplasm RFI ratios of transformants as in B. Error bars graphs correspond to the standard deviation. **D.** Drop growth assays on agar on co-transformants bearing the indicated versions of GFP-PTEN-L* and p110α-CAAX.

**Figure 6 cells-08-01512-f006:**
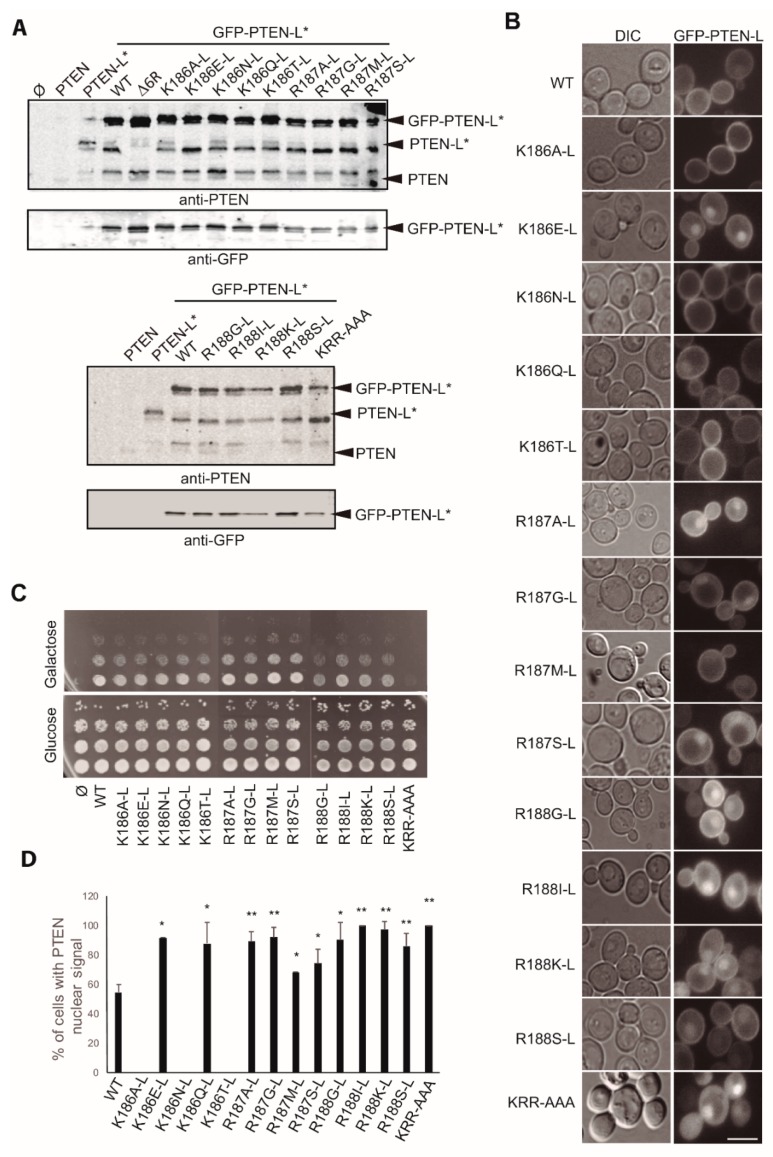
Effects of site-directed mutagenesis on the KRR NLS/PBD domains core on GFP-PTEN-L*. **A.** Immunoblots on lysates obtained from yeast transformants expressing the indicated versions of PTEN or GFP-PTEN-L*, as indicated. The same membrane was hybridized with anti-PTEN (upper panels) or anti-GFP antibodies (lower panels). **B.** Bright field (left panels) and fluorescence microscopy (right panels) of YPH499 strain transformants expressing WT GFP-PTEN-L* or the mutants indicated. Scale bar corresponds to 5 µm. **C.** Drop growth assays on agar on co-transformants bearing p110α-CAAX and the indicated versions of GFP-PTEN-L*. **D.** Percentage of cells showing nuclear GFP-PTEN-L* signal in cultures of transformants expressing the GFP-PTEN-L* versions indicated. Error bars graphs correspond to the standard deviation (**p* < 0.05; ***p* < 0.01, according to Student’s t-test).

**Figure 7 cells-08-01512-f007:**
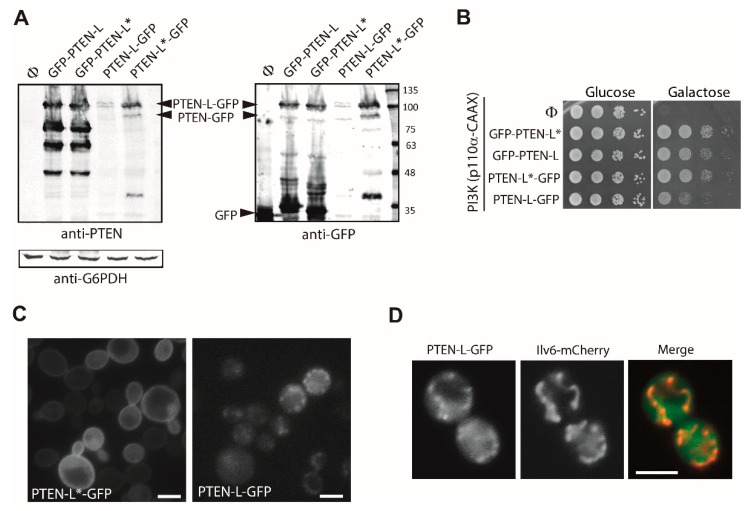
Characterization of C-terminal GFP fusions to PTEN-L in yeast. **A.** Immunoblots on lysates from YPH499 strain yeast transformants expressing the indicated versions of PTEN or GFP-PTEN and PTEN-GFP for comparison. The same membrane was hybridized with anti-PTEN antibodies (left), anti-GFP antibodies (right) and anti G6PDH (lower left panel) as loading control. **B.** Drop growth assays on agar on co-transformants expressing p110α-CAAX and the indicated versions of GFP-PTEN-L or PTEN-L-GFP for comparison. **C.** Fluorescence microscopy of representative transformants as in A-B, expressing the indicated versions of PTEN-L-GFP. **D.** Co-localization of PTEN-L-GFP with yeast mitochondria. Representative cell co-transformed with pYES2-PTEN-L-GFP and YEplac112-Ilv6-mCherry. Bars correspond to 5 µm.

**Figure 8 cells-08-01512-f008:**
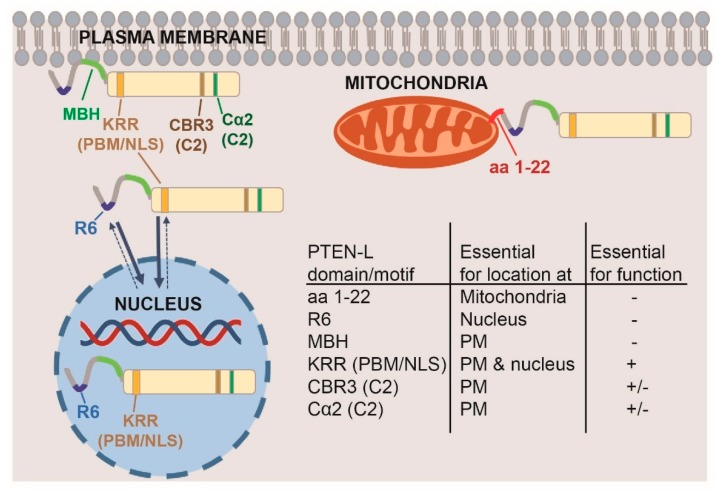
Summary of the results obtained in the yeast model on the involvement of different domains of PTEN-L in its function and localization. The N-terminal 22 amino acids of PTEN-L drive mitochondrial localization. It their absence, PTEN-L localizes to the PM (involving the MBH, PIP_2_-binding motif (PBM) and C2 domains) and the nucleus (involving the R6 motif and the KRR motif within the PBM). Only the whole KRR motif is crucial for function, while elimination of basic residues in the CBR and Cα2 loops at the C2 domain, partially alters PTEN-L activity.

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
