# Peer review of "Expression of Human PTEN-L in a Yeast Heterologous Model Unveils Specific N-Terminal Motifs Controlling PTEN-L Subcellular Localization and Function"

_cells, 2019, doi:10.3390/cells8121512_

Round 1
Reviewer 1 Report
Fernandez-Acero et al use a yeast model to explore the activity and localisation of N-terminally extended forms of PTEN. Through mutation of various regions, they highlight parts of the protein responsible for its localisation and function. The study is generally well carried out, though would benefit from clarification in a few places as detailed below.
Major Points
Survival in galactose media is much better in Figure 3A than Figure 2B, despite (from my reading) being a similar experimental set up. In addition, some PTEN constructs seem to be expressed in both Figure 3A and 3B, but growth rescue is better in Figure 3A. Can the authors explain the discrepancy? Were their results generally reproducible? In Figure 1, when the M174I mutant was used, the lowest MW PTEN band disappeared due to loss of the Met start codon for canonical PTEN. However, in Figure 5A there still appear to be 2 bands at the lowest molecular weights for the M174I mutants. Can the authors explain this? Is it due to the resolution of the gel, and if so, better resolution is needed to show this. In Figure 6B the authors say that the KRR-AAA mutant loses PM localisation. However, there is no graph to support this, and the image shows that allow more diffuse, some cells still seem to have PM localisation. Similarly, the authors state that the KRR-AAA mutant has the most pronounced nuclear localisation, but from the images, it doesn’t look much more pronounced than some of the other single residue mutations e.g. R188I-L. Graphs similar to those used in Figure 4 would be helpful here.Minor Points
The reference to Figure 2D in Results section 3.4 when describing the LLL-AAA and deltaMBH mutants seems incorrect. Figure 2D has the longer, but unmutated forms only. When describing Figure 4H, use doubled (rather than duplicated) and tripled (rather than triplicated). Figure 5A, can the authors label the major degradation band they are referring to which is missing in the R6 mutant, as none appear to be missing (just reduced). The authors state that the presence of mitochondrial spots of the PTEN-L form suggests the presence of a mitochondrial targeting sequence in the first 22 aas of PTEN-L. Is there any putative MTS in the sequence?Author Response
Major Points
-Survival in galactose media is much better in Figure 3A than Figure 2B, despite (from my reading) being a similar experimental set up. In addition, some PTEN constructs seem to be expressed in both Figure 3A and 3B, but growth rescue is better in Figure 3A. Can the authors explain the discrepancy? Were their results generally reproducible?
Yes, we agree that the plate shown in 3A is slightly overgrown as compared to others, but indeed results were reproducible beyond the subtle variability in growth efficiency that different experiments may yield. Each growth experiment must be compared to its internal positive control (PTEN-L*), as pictures may have been taken with a few hours of difference after 3 days of incubation in galactose. The results shown are representative of biological triplicates and the trend is the same in the results shown in 2B, 3A and 3B figures regardless of slight growth differences. We have improved the contrast in Fig. 3B for better comparative assessment.
-In Figure 1, when the M174I mutant was used, the lowest MW PTEN band disappeared due to loss of the Met start codon for canonical PTEN. However, in Figure 5A there still appear to be 2 bands at the lowest molecular weights for the M174I mutants. Can the authors explain this? Is it due to the resolution of the gel, and if so, better resolution is needed to show this.
The referee means comparison between Fig. 2C and 5A. Indeed the blot in Fig 5A has less resolution for the PTEN band than the one in Fig 2C. The intensity of the band corresponding to PTEN is lower than that of a close degradation band of PTEN-L. Still, even in Fig 5C it is evident that the faint band corresponding to the precise size of PTEN is missing in the PTEN-L * M174I-L, GFP-PTEN-L* M174I-L and GFP-PTEN-L*(DR6) lanes. In the last lane, all bands are shifted to a lower Mw due to the deletion.
-In Figure 6B the authors say that the KRR-AAA mutant loses PM localisation. However, there is no graph to support this, and the image shows that allow more diffuse, some cells still seem to have PM localisation. Similarly, the authors state that the KRR-AAA mutant has the most pronounced nuclear localisation, but from the images, it doesn’t look much more pronounced than some of the other single residue mutations e.g. R188I-L. Graphs similar to those used in Figure 4 would be helpful here.
We initially considered including a graph similar to those in Fig 4 also in this figure 6. However, as shown below in the graph in blue, no significant differences in PM/cytoplasm RFI ratio were recorded as compared to the wild type for any mutant, except for the triple KRR-AAA mutant, in which there is absolutely no distinction between the cytoplasmic signal and the periphery. Thus, we could not measure this ratio, because we cannot detect PM enrichment in any cell (n=300; see representative field below). In summary, we do not think that this graph adds crucial information.
We agree with the referee that the enhanced nuclear localization of the KRR-AAA mutant is not significantly more pronounced than other mutants, such as R188I-L. We have modified the text accordingly, by changing the statement “Such effect was most clear in the triple mutant KRR-AAA” to “Such effect was also observed in the triple mutant KRR-AAA”.
|
Minor Points
The reference to Figure 2D in Results section 3.4 when describing the LLL-AAA and deltaMBH mutants seems incorrect. Figure 2D has the longer, but unmutated forms only.
This was indeed wrongly referred. Fig. 4F-H was obviously meant here. This has been corrected. We thank the reviewer for detecting this mistake.
When describing Figure 4H, use doubled (rather than duplicated) and tripled (rather than triplicated).
This has been corrected.
Figure 5A, can the authors label the major degradation band they are referring to which is missing in the R6 mutant, as none appear to be missing (just reduced).
Yes, this is a reproducible result (see also Fig. 6A) and we believe it is clear. We have labeled in the figure the band that we mean and added information for interpretation to the Figure legend.
The authors state that the presence of mitochondrial spots of the PTEN-L form suggests the presence of a mitochondrial targeting sequence in the first 22 aas of PTEN-L. Is there any putative MTS in the sequence?
We have carried out bioinformatics analyses with both the MitoFates and TargetP 2.0 softwares but we have not detected any canonical MTS in the sequence, thus we do not discuss this point in the text.

Reviewer 2 Report
The manuscript by Fernandez-Acero et al. uses an elegant humanized yeast model, which they previously developed for canonical PTEN, to investigate the functional effects of molecular determinants contained within recently described N-terminally extended translational PTEN variants. In addition, they use GFP-tagged forms of the variants to determine effects of molecular determinants on subcellular localization. I think this is a carefully performed study, using multiple different controls, to provide insight into how molecular determinants contained within the PTEN variants affect their localization/function, and will be of interest for researchers working in the field.
Minor points
I suggest to add (i) a schematic describing the yeast system on the first usage (Fig. 2) to facilitate reading for the "nonaficionado", and (ii) either a diagram or table summarizing the findings of all the molecular determinants (MBH, poly-Arg, CBR3, Ca2, PIP2-BM, N-terminal amino acids) on subcellular localization/function at the end of the study.
Section 3.3, p.9 line 12. Replace "neat" with "robust"
p.14 line 5. Replace "position is a hinge" with "position can act as a pivot"
Author Response
Referee 2
Minor points
I suggest to add (i) a schematic describing the yeast system on the first usage (Fig. 2) to facilitate reading for the “nonaficionado”, and (ii) either a diagram or table summarizing the findings of all the molecular determinants (MBH, poly-Arg, CBR3, Ca2, PIP2-BM, N-terminal amino acids) on subcellular localization/function at the end of the study.
We have added a new panel in Fig. 2B explaining the rational of yeast growth-based functional analyses. Also, we have added new Fig. 8, which could also be used as a graphical abstract, to attend to this referee’s demand for a summary figure. We thank the referee for these suggestions, which we think improve the quality and clarity of our article.
Section 3.3, p.9 line 12. Replace “neat” with “robust”
Done
p.14 line 5. Replace “position is a hinge”; with “position can act as a pivot”
Done.
